# Diagnostic performance of optical coherence tomography macular ganglion cell inner plexiform layer and retinal nerve fiber layer thickness in glaucoma suspect and early glaucoma patients at St. Paul's hospital millennium medical college, Addis Ababa, Ethiopia

**Addishiwot Abera**⊕*, **Girum W. Gessesse**

Department of Ophthalmology, St Paul's Hospital Millennium Medical College, Addis Ababa, Ethiopia

\* addishiwotabera@gmail.com

## Abstract

### Purpose

To evaluate glaucoma diagnostic performance of ganglion cell inner plexiform layer and retinal nerve fiber layer parameters measured with cirrus HD optical coherence tomography (OCT).

### Method

Total of 188 eyes were included in our study. 49 eyes of healthy controls, 70 glaucoma suspect eyes and 69 early glaucomatous eyes. Complete ophthalmic examination was done including visual field test (with Humphrey field analyzer) and OCT scanning of ganglion cell inner plexiform layer (GCIPL) and retinal nerve fiber layer (RNFL) in different quadrants. Sensitivity, specificity and area under the receiver operating characteristic curve (AUROC) of each parameter was calculated to provide diagnostic ability between normal controls, glaucoma suspects or early glaucoma.

### Result

GCIPL and RNFL parameters had strong power in discriminating early glaucoma from healthy controls with all having AUROC of above 0.76. Of all the GCIPL and RNFL parameters, the only variable that could discriminate between glaucoma suspect and healthy controls was the combined parameter by OR-logic approach. Of all the parameters, the average and nasal RNFL parameters had the strongest power in discriminating between the two with AUROC of 0.81. All parameters had an overall good diagnostic performance with excellent sensitivity but the specificity was relatively poor. The combined parameter had the best specificity (62.2%) with excellent sensitivity (93.5%).

**Data Availability Statement:** All relevant data are within the paper and its Supporting Information files.

**Funding:** The authors received no specific funding for this work.

**Competing interests:** The authors have declared that no competing interests exist.

## Conclusion

Nasal RNFL parameters had the strongest power in discriminating between glaucoma suspect and healthy controls and the OR-logic combination of RNFL and GCIPL provides better diagnostic performance than single GCIPL, RNFL or ONH parameter.

## 1. Introduction

Glaucoma is a group of diseases that is characterized by optic neuropathy with excavation and undermining of the neural and connective tissue elements of the optic disc with eventual development of distinctive patterns of visual dysfunction [1]. Using data collected predominantly in 2002, World health organization (WHO) estimated blindness prevalence for all types of glaucoma was estimated as more than37 million people, placing it as the second leading cause of blindness worldwide, which previously was estimated to be third cause of blindness. However, glaucoma puts greater public health challenge than cataract because it causes irreversible blindness. More than 82% of all blind people are 50 years and older which shows as the population grows older, the prevalence rises [2].

Among white aged 40 years and older a prevalence of between 1.1% & 2.1% has been reported based on studies performed throughout the world. The prevalence among black and Latin persons is up to 4 times higher to the prevalence among whites. Black individuals are also at greater risk with increasing age: in persons 46-65years, the likelihood of blindness from glaucoma is 15 times higher among blacks than among whites [3].

Detectable glaucomatous damage to the RNFL (which is formed by the expansion of the fibers of the optic nerve) on visual field can delay by up to 5years. OCT gives a highly objective optic nerve, RNFL and macula assessment when compared to standard automated perimetry which is a subjective test with greater variability [4]. Glaucoma affects the inner most retina layers (nerve fiber layer, ganglion cell and inner plexiform layer) which we call ganglion cell complex. Eventually it causes an irreversible loss of ganglion cell axons. Given that approximately half of retinal ganglion cells are located within 4.5mm of foveal centre and we cannot detect patients at least 40% of retinal ganglion cell loss occurs, early detection of macular ganglion cell complex loss can be used for early diagnosis or following disease progression [5]. The staggering number of patients with glaucoma may be even an underestimate, since visual field loss is required in the definition of glaucoma & many individuals have glaucoma without documented visual field loss. So, early detection is mandatory to apply treatment which helps to stop or delay progression [6].

Diagnostic error accounts more when compared to medication error or failure to monitor patients. Detection of early loss of retinal nerve fiber either clinically or by means of red-free photo is also subjective and may be inconclusive. There is a risk for being inaccurate in clinical detection of suspicious optic nerve head. Looking for proof of disease progression to confirm glaucoma suspect as having pre perimetric glaucoma is risky [7].

## 2. Materials and methods

Subjects recruited in our study were from eye department of Saint Paul's hospital millennium medical college (SPHMMC), Addis Ababa, Ethiopia. This hospital was built by Emperor Haile Selassie. This hospital hosts annual average of 200,000 patients [8]. It is a cross-sectional clinic based comparative study. All consecutive patients from the three groups of study population

fulfilling the inclusion criteria were included till the sample size is achieved from January 2018- May 2019.

A total of 188 eyes (69 early glaucomatous eyes,70 glaucoma suspect eyes and 49 eyes of healthy controls) were included. Sample size was calculated by a formula for comparative study [9].

f(α,β) x 2 x SD2/ (d)2

Where f(α,β) = 7.85 for 80% (1-β) with 5% or 0.05 significance (α)

SD- Standard Deviation of the outcome of interest. It was taken from a similar study [10].

d- The smallest difference in means that it would be clinically meaningful to detect. It was taken as 5.

The sample size needed for each group were 70

Complete ophthalmic examination was done for all subjects: past and recent ocular and medical history, visual acuity test with Snellen E chart, non-contact tonometry, gonioscopy, slit lamp examination with 90 dioptre volk lens. Visual field exam was also taken for each eye using standard automated perimetry done with a Humphrey field analyzer (Carl Zeiss Meditec Inc) program 24-2and papillary and macular imaging using Cirrus HD-OCT (Carl Zeiss Meditec's proprietary software).

## 2.1. Inclusion and exclusion criteria

The inclusion criteria for all the groups was age ≥ 18 years, best corrected visual acuity ≥ 6/18 with spherical equivalent between -5.00 & +5.00 dioptre & a cylinder correction with ± 2.00 dioptre and open anterior chamber angle. The healthy control subjects had normal optic nerve head (ONH) which is the point of exit for ganglion cell axon, intraocular pressure of ≤21 mmhg, a normal reliable SITA 24–2 visual field with Humphrey field analyzer. Glaucoma suspects had one of optic nerve head changes (asymmetric cup to disc ratio > 0.2 unexplained by the ONH size difference (the white cup is a pit with no nerve fibers), enlarged cup to disc ratio>0.6, disc haemorrhage, suspicious alteration in the nerve fiber layer, narrowing or notching of neuroretinal rim), normal mean deviation (MD 0dB to -2dB), normal pattern standard deviation (PSD with absence of a point with p<5%) and normal glaucoma hemi field test on perimetry (1). Early glaucomatous eyes had one of the ONH changes mentioned above with cup to disc ratio≤0.65 and/or mild visual field defect not within 10 degrees of fixation and nerve fiber layer defects which goes with visual field changes [1].

Exclusion criteria for all three groups was media opacity which compromises image quality (corneal opacity, cataract), any retinopathy that can cause visual field or optic disc abnormalities, macular pathology (age related macular degeneration), history of chronic ocular or systemic steroid use or cerebro-vascular event that could affect VFT, high Refractive error (±5dioptre), OCT image quality<6, any history of intraocular surgery, angle closure, signs suggesting non glaucomatous optic neuropathy (remaining neuroretinal rim pallor, visual field defect respecting the vertical midline), secondary glaucoma (uveitic glaucoma, previous trauma).

## 2.2. OCT imaging procedure

Macular GCIPL and Peri-papillary RNFL of each participant was taken by optometrists after pupil dialated with 1% tropicamide. GCIPL scan with Cirrus HD-OCT was used to scan the macula over a7mm×6mm region by using macular cube 512x128 scan mode and the peri papillary RNFL thickness was measured in the optic disc cube 200x200 scan mode, which scan a 6x6mm square centered on the ONH. The result was interpreted in a colour coded manner. If it is colored as green, the probability value is within 5%–95%, and labelled as 'within normal

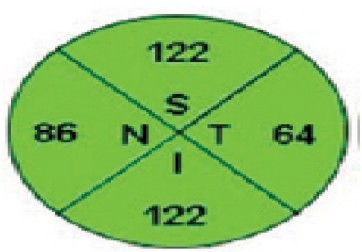 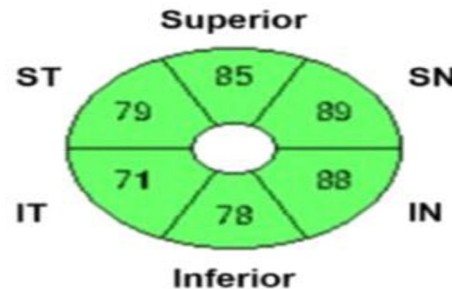

**Fig 1. Representative OCT image of peripapillary RNFL in four quadrants (left) & macular GCIPL in six quadrants (right).** N- nasal, S- superior, T-temporal, I- inferior, ST- superotemporal, SN- superonasal, IT- inferotemporal & IN-inferonasal.

limit'. If it is colored as yellow, the probability value is less than 5% but greater than1%, and labelled as 'borderline'. If it is colored as red, the probability value is less than1%, and labelled as 'outside normal limit'. The scan was taken as abnormal when it is flagged as red. Scans with segmentation error, motion artefact and quality of <6 were excluded. Below is a representative OCT image of peripapillary RNFL in four quadrants (superior, inferior, nasal and temporal) & macular GCIPL in six quadrant (superior, superonasal, superotemporal, inferior, inferonasal & inferotemporal) respectively. See "Fig 1".

### 2.3. Statistical analysis

It was performed with SPSS version 23.0. The Or-logic approach was used. This approach requires at least one of the parameters (Either RNFL OR GCIPL) to be abnormal so as to be suggestive of glaucoma. Unlike And-logic approach which requires both parameters be flagged as red to be taken glaucomatous, the Or-logic approach will use either RNFL or GCIPL red flagged result to be glaucomatous. Examination of ONH with 90D volk lens was taken as gold standard way to calculate sensitivity and specificity. The chi-square test was used to compare specificities and sensitivities. The independent sample t test was also used to determine and compare variables between groups. Receiver operating characteristic (ROC) curve showed the relation between sensitivity and 1-specificity for a given diagnostic test. AUROC which is a graphical plot to illustrate diagnostic ability has been used to assess the discriminating ability between groups. An AUROC of 1.0 represents perfect discrimination, where as an AUROC of 0.5 represents chance discrimination. P value of less than 0.05 was used to indicate a statistically significant difference.

### 2.4. Ethical consideration

We conducted the study after acquiring an ethical clearance letter from Saint Paul's hospital IRB committee. The name of the study participants was not registered. Informed consent was taken from the participants after assuring them that they could refrain from the interview if they are not willing.

## 3. Result

A total of 188 eyes were involved in our study including early glaucoma 36.7%, glaucoma suspects 37.2% and healthy control 26.1%. The mean age of participants for early glaucoma patients was 58years (SD±10), for glaucoma suspects 49years (SD±12) and for healthy controls 44 years (SD±12). Male proportion was 52.1% & female 47.9%. Hypertension was present

**Table 1. Demographic characteristics of participants.**

| Parameters | Early glaucoma | Glaucoma suspect | Healthy control |
|---|---|---|---|
| Participant no. | 69(36.7%) | 70(37.2%) | 49(26.1%) |
| Age (mean ± SD) | 58±10 | 49±12 | 44±12 |
| Sex (Male/Female) | 44/25 | 31/39 | 23/26 |
| IOP (mmhg) | 20.6±5.6 | 19.1±5.3 | 16±3.9 |
| MD (dB) | -5.9±3.5 | -3.8±1.7 | -3.4±1.8 |
| PSD | 3.2±2.5 | 2.1±0.9 | 1.8±0.7 |
| VCDR | 0.61±0.1 | 0.64±0.16 | 0.59±0.1 |
| Rim area (μm) | 1.21±0.2 | 1.21±0.2 | 1.29±0.2 |

SD- standard deviation, IOP- intraocular pressure, MD- mean deviation, PSD- pattern standard deviation, VCDR- vertical cup to disc ratio

among 22.3% & diabetes among 18.1%. Family history of glaucoma was present in only 6 patients (3.2%) and no participant was taking steroid "Table 1".

On comparison of GCIPL parameters among glaucoma suspects and healthy controls, all GCIPL quadrant thicknesses were significantly different except superior and temporal quadrant. In contrast to the observed large differences in GCIPL parameters among early glaucoma and controls, the differences were less marked among glaucoma suspects and controls. All RNFL parameters were different significantly in these groups. Compared to GCIPL parameters, the observed differences were larger in RNFL parameters (22.3) among suspects and controls. Of the RNFL parameters, the largest difference was observed on the inferior quadrant "Tables 2 and 3".

The overall diagnostic accuracy of combined and individual GCIPL and RNFL parameters were tested by receiver operating curve (ROC). For detection of early glaucoma from healthy controls, the combined or logic (that requires either RNFL OR GCIPL to be abnormal) and all of the GCIPL parameters had good diagnostic performance with all having AUROC above 0.7. Among the RNFL parameters, superior and temporal thickness had poor performance (0.682, 0.647 respectively) while the average, inferior and nasal thickness had good diagnostic

**Table 2. Mean ± standard deviation of GCIPL and RNFL parameters.**

| Parameter | Mean ± SD | | |
|---|---|---|---|
| | Early glaucoma | Healthy control | Glaucoma suspect |
| GCIPL Average | 74.2μm ±12.4 | 83.4μm ±5.7 | 79.8μm±10.9 |
| GCIPL Minimum | 67.2μm ±16.2 | 79.9μm ±6.6 | 74.9μm±14.5 |
| GCIPL Superior | 74.9μm ±14.7 | 84μm ±6.3 | 80.9μm±11.1 |
| GCIPL Inferior | 72.7μm ±13.7 | 81.7μm ±6.7 | 77.5μm±11.5 |
| GCIPL Superonasal | 76.5μm ±14.4 | 85.2μm ±6.8 | 81.5μm±11.2 |
| GCIPL Inferonasal | 74.7μm ±13.5 | 84.1μm ±7.0 | 80.3μm±12.0 |
| GCIPL Superotemporal | 73.6μm ±13.2 | 82.8μm ±5.3 | 79.5μm±10.8 |
| GCIPL Inferotemporal | 73.2μm ±12.9 | 82.5μm ±7.1 | 79.1μm±12.1 |
| RNFL Average | 84.0μm ±7.35 | 97.8μm ±9.3 | 92.3μm ±9.9 |
| RNFL Superior | 101.8μm ±8.41 | 121.6μm±15.5 | 114.9μm±17.2 |
| RNFL Inferior | 105.7μm ±5.36 | 128.0μm±16.7 | 118.3μm±16.8 |
| RNFL Nasal | 70.1μm ±8.84 | 78.2μm ±12.0 | 73.8μm±10.5 |
| RNFL temporal | 57.8μm ±5.51 | 63.3μm ±8.3 | 61.3μm ±8.8 |

SD- standard deviation, GCIPL–ganglion cell inner plexiform layer, RNFL- retinal nerve fiber layer

**Table 3. The association of mean difference of GCIPL and RNFL parameters.**

| Parameter | Mean difference | | | |
|---|---|---|---|---|
| | EG Vs HC | P1 | GS Vs HC | P2 |
| GCIPL Average | -9.2 | .000 | -3.6 | .036 |
| GCIPL Minimum | -12.7 | .000 | -5.07 | .024 |
| GCIPL Superior | -9.1 | .000 | -3.1 | .086 |
| GCIPL Inferior | -9.0 | .000 | -4.2 | .024 |
| GCIPL Superonasal | -8.7 | .000 | -3.7 | .043 |
| GCIPL Inferonasal | -9.2 | .000 | -3.8 | .047 |
| GCIPL Superotemporal | -9.2 | .000 | -3.2 | .054 |
| GCIPL Inferotemporal | -9.2 | .000 | -3.4 | .079 |
| RNFL Average | -13.7 | .000 | -5.5 | .003 |
| RNFL Superior | -19.7 | .000 | -6.7 | .033 |
| RNFL Inferior | -22.3 | .000 | -9.7 | .002 |
| RNFL Nasal | -8.0 | .004 | -4.4 | .036 |
| RNFL temporal | -5.5 | .005 | -2.0 | .212 |

EG- early glaucoma, HC- healthy control, GS-glaucoma suspect, GCIPL-ganglion cell inner plexiform layer, RNFL-retinal nerve fiber layer

performance (0.819, 0.813 and 0.726 respectively). Overall, RNFL inferior and average thicknesses were the most accurate parameters in detecting early glaucoma from healthy control with AUROC of 0.81. The combined GCIPL and RNFL parameters and the GCIPL parameters individually had lower performance compared to the RNFL average and nasal thickness, but overall accuracy is still good with AUROC of 0.77 and 0.76 each. See "Fig 2".

All GCIPL and RNFL parameters had poor performance in detecting glaucoma suspects from healthy control except the combination of both parameters by or logic (sensitivity 88.5% and specificity 49.5%). The AUC for this parameter was 0.63, which had relatively better but still weak discriminating power. AUROC, specificity and sensitivity of the GCIPL and RNFL parameters for each quadrant in diagnosis of early glaucoma and glaucoma suspects is shown below "Tables 4 and 5".

There was statistically significant difference between sex only in the superior quadrant from GCIPL parameter (p = 0.02) and inferior quadrant from RNFL parameter (p = 0.027).

## 4. Discussion

It is known that structural damage be ahead of visual field loss detected by standard perimetry. So quantitative analysis of RNFL and ONH is more reliable way of glaucoma assessment [11]. Ishkawa et al measured macular sub layer thickness and showed that macular ganglion cell complex was thinner in eyes with perimetric glaucoma [12].

Leung et al used the Stratus TD-OCT, evaluated macular RNFL and report decreased RNFL thickness in glaucomatous eyes compared with normal eyes [13]. Many studies have been done to find out which OCT analysis yield the best discrimination performance.

GCIPL and RNFL measurements by using OCT has been suggested as one means of early diagnosing glaucoma patients [14–19]. In our study we assessed diagnostic performances of GCIPL and RNFL parameters individually or in combination among cases of early glaucoma, glaucoma suspects and healthy controls. The mean thickness of all parameters was significantly lower in early glaucoma and glaucoma suspects compared to healthy controls. The differences were large in early glaucoma while it is moderate in suspects compared to healthy controls.

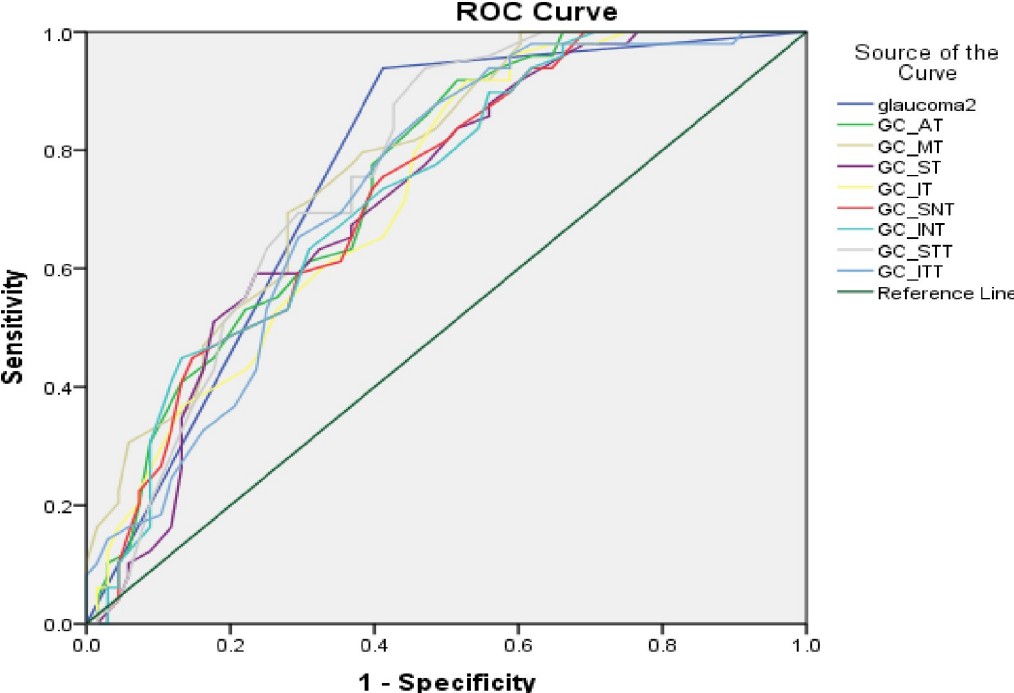

**Fig 2. Receiver operating curve of GCIPL showing healthy versus glaucoma suspect & early glaucoma.** GC-ganglion cell inner plexiform layer, AT-average thickness, MT-minimum thickness, ST-superior thickness, IT-inferior thickness, SNT-superonasal thickness, INT-inferonasal thickness, STT-superotemporal thickness, ITT-inferotemporal thickness.

The overall accuracy of each parameter was determined by using the AUROC. The AUROC assesses the overall accuracy of a diagnostic test by plotting the rate of true positive against that of false positive rate. An ideal test will have the AUROC of 1 which means 100% sensitive and specific. Practically this is rare and there will always be a trade off between sensitivity and specificity. For a test to be considered diagnostically useful, it is expected to have

**Table 4. Area under receiver operating curve (AUROC), specificity and sensitivity of the GCIPL parameters in diagnosis of early glaucoma (EG) and glaucoma suspect (GS).**

| | EG Vs HC | | | | GS Vs HC | | | |
|---|---|---|---|---|---|---|---|---|
| | AUROC | P | Sensitivity | Specificity | AUROC | P | Sensitivity | Specificity |
| **Or–logic** | 0.764 | 0.000 | 93.2% | 62.2% | 0.634 | 0.013 | 88.5% | 49.5% |
| **GCIPL-A** | 0.749 | 0.000 | 100.0% | 51.6% | 0.594 | 0.080 | 100.0% | 45.0% |
| **GCIPL-M** | 0.774 | 0.000 | 87.5% | 52.3% | 0.591 | 0.093 | 80.0% | 45.5% |
| **GCIPL-S** | 0.722 | 0.000 | 100.0% | 50.5% | 0.580 | 0.140 | 100.0% | 43.8% |
| **GCIPL-I** | 0.727 | 0.000 | 95.2% | 49.5% | 0.599 | 0.067 | 90.0% | 44.0% |
| **GCIPL-SN** | 0.729 | 0.000 | 100.0% | 50.5% | 0.579 | 0.141 | 100.0% | 45.0% |
| **GCIPL-IN** | 0.732 | 0.000 | 100.0% | 50.0% | 0.573 | 0.175 | 100.0% | 43.0% |
| **GCIPL-ST** | 0.764 | 0.000 | 100.0% | 52.1% | 0.596 | 0.076 | 100.0% | 43.8% |
| **GCIPL-IT** | 0.735 | 0.000 | 95.7% | 51.1% | 0.564 | 0.238 | 90.9% | 44.4% |

GCIPL-ganglion cell inner plexiform layer, HC- healthy control, P- P value, A–average, M- minimum, S- superior, I-inferior, SN-superonasal, IN-inferonasal, ST-superotemporal, IT-inferotemporal

**Table 5. Area under receiver operating curve (AUROC), sensitivity and specificity of the RNFL parameters in diagnosis of early glaucoma (EG) and glaucoma suspect (GS).**

|  | EG Vs HC | | | | GS Vs HC | | | |
|---|---|---|---|---|---|---|---|---|
|  | AUROC | P | Sensitivity | Specificity | AUROC | P | Sensitivity | Specificity |
| RNFL Average | 0.819 | 0.006 | 95.8% | 51.6% | 0.684 | 0.111 | 85.7% | 42.9% |
| RNFL Superior | 0.682 | 0.115 | 100.0% | 50.0% | 0.504 | 0.972 | 100.0% | 44.1% |
| RNFL Inferior | 0.813 | 0.007 | 100.0% | 50.0% | 0.715 | 0.062 | 100.0% | 42.2% |
| RNFL Nasal | 0.726 | 0.050 | 100.0% | 43.0% | 0.684 | 0.111 | 100.0% | 41.5% |
| RNFL Temporal | 0.647 | 0.202 | 90.9% | 44.9% | 0.517 | 0.880 | 83.3% | 42.5% |

P- P value, HC- healthy control, RNFL- retinal nerve fiber layer

AUROC above 0.7 and its accuracy increases as the value approaches 1. While a test with AUROC of 0.5 and lower is considered to be not better than a chance and has limited diagnostic utility [20].

In our study, specificity for most of the parameters was in the range of 40% to 60% and the maximum specificity was 62.2%. While the parameters had excellent sensitivity with most having sensitivity of 90%-100% and no parameter had sensitivity of below 70%. One study shows OR-logic combination of inferior RNFL with inferotemporal quadrant result in significantly increased sensitivity while good specificity is maintained. In contrast to our study, OR-logic combination of rim area and minimum GCIPL result in equivalent diagnostic performance to that of average RNFL and minimum GCIPL combination [21].

Another study done in Italy showed high specificity for all parameters but sensitivity was poor for eyes with preperimetric glaucoma. In this study widest AUC was recorded in RNFL OR (0.79, 95%CI 0.72–0.86) and RNFL inferior (0.76, 95%CI 0.66–0.82). Among GCIPL parameters, the best in terms of diagnostic accuracy were GCIPL OR (0.79, 95%CI 0.71–0.85) and GCIPL inferior (0.77,95% CI 0.70–0.84) [22].

In our study for glaucoma suspect, sensitivity ranges 80% -100% and specificity ranges 43% - 49.5% for GCIPL and for RNFL sensitivity ranges from 83.3% -100% and specificity from 41.5% - 44.1%. For early glaucoma sensitivity ranges 87.5% - 100% and specificity ranges 49.5% - 62.2% for GCIPL and for RNFL sensitivity ranges from 90.9% - 100% and specificity from 43.0% - 51.6%. In a study done at multi centres (Florida, Texas and California) minimum GCIPL outperformed when compared with other single GCIPL parameters with 82% sensitivity and 87.8% specificity. From single RNFL parameters, the best discriminant was the inferior quadrant with sensitivity of 74% and specificity of 95.9% followed by RNFL average. Other RNFL parameters had specificity range of 81.8% - 100% and sensitivity range of 10% - 48%.

Combining any abnormal GCIPL or RNFL parameters had sensitivity of 96% and specificities of 63.6%. Combining the best performing GCIPL or RNFL had the best performance with sensitivity of 94% & specificity of 85.7%. The sensitivity and specificity of combining any abnormal RNFL with abnormal GCIPL parameter in this study was 96% and 63.6% which was very close to our finding. The other similarity between this study and ours is the fact that the average and inferior RNFL are good parameters. On the other hand, a sharp contrast between this study and ours was observed in sensitivity which was low as 10%, and for most of the parameters specificity was very good as high as 95% [21].

In another study from Turkey, all the GCIPL parameters could discriminate early glaucoma from healthy controls. All the GCIPL parameters had the AUROC of 0.68–0.76. This is very similar to the findings in our study (AUROC 0.647–0.819). Of the RNFL parameters, except temporal quadrant, all other could discriminate early glaucoma from healthy controls. The highest of them was RNFL superior quadrant with AUROC of 0.81. In this similar study, all of the GCIPL parameters except the superotemporal quadrant could discriminate glaucoma suspects from healthy controls, while none of the RNFL parameters could discriminate the two. In contrast, except the combined parameter, none of the GCIPL or RNFL parameters could discriminate glaucoma suspects from healthy controls in our study (p>0.05). Although this study mentioned that most of GCIPL parameters could discriminate b/n the two groups, the AUROC was still weak with all in the range of 0.61–0.67.

These findings suggest the diagnostic performance of RNFL and GCIPL parameters are relatively weak in discriminating glaucoma suspects from healthy controls. This is possibly due to the blurred grey zone for normal and abnormal in the two conditions [23].

## 5. Conclusion and recommendation

GCIPL and RNFL parameters can be useful to make early diagnosis of glaucoma. Results from different studies show in consistent results on specificity and sensitivity of the individual and combined parameters. This inconsistency is due to difference in study design, the study participants and differences in measurement of the parameters. To conclude, high sensitivity means if we found a result "within normal limit" we can be sure that there is no structural damage and low specificity means if we found "outside normal limit" result we cannot correlate it with structural damage. The value of cut off point to define an abnormality should be adjusted to improve the specificity while maintaining an acceptable level of sensitivity is the recommendation from our study.

## 6. Limitation

Unmatched age between normal control & glaucoma patients and fewer number of normal controls was the limitation.

## Supporting information

**S1 File.**
(SAV)

## Acknowledgments

We would like to thank St Paul's hospital, department of ophthalmology for giving us this chance to conduct a research & Mr Tewodros for his support.

## Author Contributions

**Formal analysis:** Addishiwot Abera, Girum W. Gessesse.

**Methodology:** Addishiwot Abera, Girum W. Gessesse.

**Supervision:** Addishiwot Abera, Girum W. Gessesse.

**Validation:** Addishiwot Abera, Girum W. Gessesse.

**Writing – original draft:** Addishiwot Abera, Girum W. Gessesse.

**Writing – review & editing:** Addishiwot Abera, Girum W. Gessesse.

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
