## [Decision Letter · Decision Letter 0]

8 Oct 2021

PONE-D-21-04965Diagnostic Performance Of Optical Coherence Tomography Macular Ganglion Cell Inner Plexiform Layer And  Retinal Nerve Fiber Layer Thickness In Glaucoma Suspect And Early Glaucoma Patients At St. Paul’s Hospital Millennium Medical College, EthiopiaPLOS ONE

Dear Dr. Abera,

Thank you for submitting your manuscript to PLOS ONE. After careful consideration, we feel that it has merit but does not fully meet PLOS ONE’s publication criteria as it currently stands. Therefore, we invite you to submit a revised version of the manuscript that addresses the points raised during the review process.

We look forward to receiving your revised manuscript.

Kind regards,

Akram Belghith

Academic Editor

PLOS ONE

Journal Requirements:

2. Please update your submission to use the PLOS LaTeX template. The template and more information on our requirements for LaTeX submissions can be found at http://journals.plos.org/plosone/s/latex

Review Comments to the Author

The manuscript describes diagnostic performance of clinical available OCT system (Cirrus HD-OCT) with thickness measurement for detecting glaucoma. Both macular and peri-papillary scans were performed on 188 eyes for discrimination among early glaucoma, glaucoma suspect, and healthy control groups. Thickness of ganglion cell inner plexiform layer (GCIPL)and retinal never fiber layer (RNFL) was measured for macular scan and peri-papillary scan respectively. The study is finely designed, but the content of the manuscript is more suitable for a clinical journal. I also have major concerns about the presentation of the results and the quantification methods for data analysis, which I think the authors must address. I also offer some detailed comments to help improve the manuscript.

Major comments:

1. Missing important technical details prevents proper evaluation of the results. The authors used specificity and sensitivity to evaluate the diagnostic performance, however, how the two parameters were calculated from the layer thickness measurement was not explained. Other details about AUROC, GCIPL and RNFL parameter extraction from OCT images, OR-logic approach for analysis were missing as well. The necessary detailed information shall be provided.

2. Plots for AUROC and representative OCT images shall be presented to help the readers understand the analysis method and results. See detailed comments.

3. Segmentation errors may occur in the clinical OCT devices. Did the authors verify the segmentation accuracy?

Detailed comments:

1. Please make sure all the abbreviation are defined in the first place they appear in the main texts. Some undefined terms are: POAG, RNFL, AUROC, SD, VCDR, ONH.

2. Page 10, Study population section: please explain the meaning of the equation (Eq. xx) and what each parameter are in the formula.

3. Page 10, last paragraph: Please clarify the threshold for normal MD and PSD.

4. Page 10, last paragraph: Please provide reference for the disease classification.

5. Page 11, OCT imaging procedure section: Please explain how the OCT thickness was extracted for different quadrants, and how the quadrants are defined. A representative OCT image should be provided for clarity purpose to help the readers understand the data analysis strategy.

6. Page 11, first paragraph: The unit is missing for refractive error.

7. Page 11, Statistical analysis section: Please explain the ‘Or-logic’ approach.

8. Page 11, Statistical analysis section: Please clarify how ‘specificities’ and ‘sensitivities’ are calculated.

9. Page 13, Result section: Are the enrolled groups age-matched? Inner retinal thickness also decreases with aging.

10. Page 14 table 2: It is not clear how the GCIPL and RNFL parameters are extracted at different regions. Example of OCT image shall be provided to explain the methods and data analysis strategy.

11. Page 14 table 2: Units are missing for the parameters.

12. Page 14 paragraph after table 2: ROC plots shall be provided for data visualization and evaluation.

13. Page 14 paragraph after table 2: Please explain what ‘or logic’ is and how ‘or logic’ is determined and used in your analysis.

14. Page 16 second paragraph: Please define ‘b/n’.

15. Page 18, Discussion section, the first sentence. I believe it is still a debatable question what loss first in glaucoma.

---

## [Editor Report · Decision Letter 1]

2 Feb 2022

Diagnostic performance of optical coherence tomography macular ganglion cell inner plexiform layer&retinal nerve fiber layer thickness in glaucoma suspect&early glaucoma patients at St. Paul’s hospital millennium medical college, Addis Ababa,Ethiopia

PONE-D-21-04965R1

Dear Dr. Abera,

We’re pleased to inform you that your manuscript has been judged scientifically suitable for publication and will be formally accepted for publication once it meets all outstanding technical requirements.

Kind regards,

Akram Belghith

Academic Editor

PLOS ONE

---

## [Editor Report · Acceptance letter]

1 Jun 2022

PONE-D-21-04965R1 

Diagnostic performance of optical coherence tomography macular ganglion cell inner plexiform layer&retinal nerve fiber layer thickness in glaucoma suspect&early glaucoma patients at St. Paul’s hospital millennium medical college, Addis Ababa,Ethiopia 

Dear Dr. Abera:

I'm pleased to inform you that your manuscript has been deemed suitable for publication in PLOS ONE. Congratulations! Your manuscript is now with our production department. 

Kind regards, 

on behalf of

Dr. Akram Belghith 

Academic Editor

PLOS ONE